# Effect of Ni Substitution on Thermoelectric Properties of Bulk *β*-Fe_1−x_Ni_x_Si_2_ (0 ≤ x ≤ 0.03)

**DOI:** 10.3390/ma16030927

**Published:** 2023-01-18

**Authors:** Sopheap Sam, Soma Odagawa, Hiroshi Nakatsugawa, Yoichi Okamoto

**Affiliations:** 1Yokohama National University, Yokohama 240-8501, Japan; 2National Defense Academy, Yokosuka 239-8686, Japan

**Keywords:** iron silicide, bipolar effect, Ni doping, thermoelectric properties, *ZT* values

## Abstract

A thermoelectric generator, as a solid-state device, is considered a potential candidate for recovering waste heat directly as electrical energy without any moving parts. However, thermoelectric materials limit the application of thermoelectric devices due to their high costs. Therefore, in this work, we attempt to improve the thermoelectric properties of a low-cost material, iron silicide, by optimizing the Ni doping level. The influence of Ni substitution on the structure and electrical and thermoelectric characteristics of bulk *β*-Fe_x_Ni_1−x_Si_2_ (0 ≤ x ≤ 0.03) prepared by the conventional arc-melting method is investigated. The thermoelectric properties are reported over the temperature range of 80–800 K. At high temperatures, the Seebeck coefficients of Ni-substituted materials are higher and more uniform than that of the pristine material as a result of the reduced bipolar effect. The electrical resistivity decreases with increasing x owing to the increases in metallic ε-phase and carrier density. The ε-phase increases with Ni substitution, and solid solution limits of Ni in *β*-FeSi_2_ can be lower than 1%. The highest power factor of 200 μWm^−1^K^−2^ at 600 K is obtained for x = 0.001, resulting in the enhanced *ZT* value of 0.019 at 600 K.

## 1. Introduction

Thermoelectricity has been considered a potential technique to recover waste heat into electrical energy through the Seebeck effect without exhaust gas pollution to the environment, with no moving parts, and with no necessary maintenance required. To achieve highly efficient thermoelectric (TE) devices, finding promising semiconducting materials is the main challenge. Traditional TE materials such as PbTe and Bi_2_Te_3_ are high-priced and toxic; therefore, researchers have been trying to develop abundant and non-toxic materials, such as binary copper chalcogenide [1], copper sulfide compound [2,3], iron silicide [4,5,6,7,8], and other materials, in order to replace those traditional ones. Iron silicide compound is an abundant and non-toxic material having three different kinds of phases, such as the cubic *ε*-phase with space group *P2*_1_*3* [4,5], the tetragonal *α*-phase with space group *P4/mmm* [6,7], and the orthorhombic *β*-phase with space group *Cmce* [8]. According to Piton and Fay diagram [9], the semiconducting *β*-phase can be formed at a temperature below 1259 K and depends on the kind and amount of external impurity doping, whereas the metallic ε and α-phases are grown at a higher temperature. It is noticed that its ε and α-phases are a metal that is not suitable for TE applications due to the deterioration of the Seebeck coefficient (*S =* − ∆*V*/∆*T,* where ∆*V* and ∆*T* are the TE voltage and temperature difference across the material, respectively). However, its β-phase, a semiconducting material with a small band gap of around 0.7 eV [10], is suitable in TE applications. In addition, compared to other traditional TE materials (PbTe and Bi_2_Te_3_), *β*-FeSi_2_ can work at high temperatures due to strong oxidation resistance, good thermal stability, and low cost [11,12,13,14,15]. However, due to its narrow band gap and low carrier concentration (*n*_H_) of around 10^16^ cm^−3^, the bipolar effect usually occurs in a non-doped *β*-FeSi_2_, especially in high-temperature regions, resulting in a decline of the |*S*|. The Seebeck effect is generated by two types of carriers having opposite signs. With the increased temperature and low *n*_H_, the total Seebeck effect is cancelled out, which is unfavorable for TE applications [16,17,18]. Therefore, as temperature increases, the TE performance is defined by *ZT* = *S*^2^*ρ*^−1^*κ*^−1^*T*, where *S*, *ρ*, *κ*, and *T* are Seebeck coefficient, electrical resistivity, total thermal conductivity, and temperature, respectively. The Seebeck coefficient worsens due to the decrease in |*S*| caused by the bipolar effect. The pristine *β*-FeSi_2_ has a low value of *ZT* of only round 2 × 10^−4^ [19]. To solve this issue, doping with impurities having a large valence electron in either Fe or Si sites is considered an effective technique for increasing the *n*_H_, resulting in an improvement in the stability of |*S*| [17]. In addition, the *ρ* is inversely proportional to the *n*_H_; therefore, it can be simultaneously decreased owing to the increase in *n*_H_. As a result, the *ZT* can be significantly improved due to the monotonicity in |*S*| and the decrease in *ρ*. Theoretically, it was reported that the optimum *n*_H_ to improve the TE performance of *β*-FeSi_2_ is approximately within the range of 1 × 10^20^ to 2 × 10^21^ cm^−3^ [15]. In fact, we prepared *β*-Fe_0.97_Co_0.03_Si_2_ with the arc melting method and found a *ZT* value of 0.099 at 800 K [20]. Furthermore, many previous works [21,22,23,24,25,26,27,28,29,30] attempted to enhance the *n*_H_ of *β*-FeSi_2_ via doping with various impurities.

Ito et al. reported the TE characteristics of *β*-FeSi_2−x_P_x_ fabricated by mechanical alloyed (MA) and hot-pressed (HP) method. By doping P at the Si site, the *S* was negative, indicating the n-type material with the optimum concentration of x = 0.02, and the *ρ* slightly decreased with a considerable increase in P content due to the increase in *n*_H_. As a result, the highest *ZT* of about 0.033 at 672 K was obtained with x = 0.02, which was about 11 times higher than that of the non-doped sample [21]. In addition, Tani and Kido found that the *ZT* of *β*-FeSi_2_ can be enhanced up to 0.14 at 847 K by doping with Pt as an impurity [22]. Ohtaki et al. investigated various impurities for doping, such as Cu, Zn, Nb, Ag, Sb, and Mn, by analyzing the microstructural changes and TE performance of *β*-FeSi_2_. They reported that the microstructures were remarkably changed by those impurities. The highest *ZT* value of about 0.026 was obtained by 3% Mn doping at 873 K [23]. In addition, Chen et al. investigated the thermoelectric characteristics of Co addition on *β*-FeSi_2_ fabricated by rapid solidification and followed the HP method. It was reported that the optimum doping to achieve maximum *ZT* = 0.25 was obtained in Fe_0.94_Co_0.06_Si_2_ samples due to the enhancement in *S* and a significant reduction in *ρ* [24]. Furthermore, Du et al. attempted to improve the TE performance of the previous Fe_0.94_Co_0.06_Si_2_ by doping with an additional impurity element named Ru. It was found that Ru doping significantly decreases the thermal conductivity because the strain field and mass oscillation scatter the phonons, resulting in the improvement of *ZT* = 0.33 at 900 K [25]. Moreover, Dabrowski et al. investigated the effects of several dopants, namely, Mn, Co, Al, and P, on the TE properties of *β*-FeSi_2_. They reported that compared to other dopants, P was not effective at improving the *ZT* due to only a slight decrease in *ρ*, where the *n*_H_ of the P-doped sample was probably lower than that of other impurity-doped samples; however, the highest *ZT* was obtained in a Co-doped sample, probably due to the high *n*_H_ [26]. Qiu et al. have recently reported that by doping 16% Ir into the Fe site of *β*-FeSi_2_, the *ZT* can be greatly improved to 0.6 at 1000 K due to the significant reductions in *ρ* and *κ*, resulting from high *n*_H_ and phonon–electron scattering, respectively [27]. Based on a series of previous reports, it is worth noticing that doping with elements having large valence electrons to either Fe or Si sites of *β*-FeSi_2_-based materials is remarkably effective at improving the *n*_H_ and the TE performance.

Since Ni has two valence electrons more than Fe, the *n*_H_ can be possibly increased by substituting Ni into the Fe site of *β*-FeSi_2_. Komabayashi et al. reported that the *S, ρ*, and power factor (*PF* = *S*^2^*ρ*^−1^) at room temperature of Fe_0.94_Ni_0.06_Si_2.05_ thin film fabricated by the RF sputtering method were −113 μVK^−1^, 0.076 Ωcm, and 17 μWm^−1^K^−2^, respectively [28]. In addition, Nagai et al. investigated the effect of Ni addition on the *PF* of *β*-FeSi_2_ fabricated by mechanical alloying and hot-pressing techniques. The highest |*S*| was obtained after 1% Ni doping—240 μVK^−1^ at 600 K—and *ρ* significantly decreased with Ni addition. This indicates that both *S* and *ρ* can be simultaneously improved by Ni addition, contributing to the enhancement of *PF* [29]. Furthermore, Tani and Kido reported that the *ρ* of bulk *β*-Fe_1−x_Ni_x_Si_2_ decreased with the substitution of Ni owing to the increase in *n*_H_ [30]; thus, a reduction in the bipolar effect should be achieved. However, there are only a few reports regarding the effect of Ni doping of *β*-FeSi_2_, and an investigation on the thermal conductivity (*κ*) and the *ZT* values has not been reported. Moreover, the optimum Ni doping concentration needed to improve the TE performance of *β*-Fe_1−x_Ni_x_Si_2_ also has not been investigated yet.

In this work, we attempted to improve the electrical and thermoelectric properties of the bulk of binary *β*-FeSi_2_ by Ni substitution into the Fe site prepared by the facile arc-melting techniques and directly followed by a heat treatment and annealing process. For the *β*-Fe_1−x_Ni_x_Si_2_ system (0 ≤ x ≤ 0.03), a detailed investigation of the optimum doping level of Ni to enhance TE performance is reported for the temperature range of 80–800 K.

## 2. Materials and Methods

### 2.1. Sample Fabrication

The raw materials of Fe grain (99.9% up, 3Nup, High Purity Chemicals, Japan), Si grain (99.999%, 5N, High Purity Chemicals, Japan), and Ni grain (99.9%, 3N, High Purity Chemicals, Japan) were prepared following the composition of Fe_x_Ni_1−x_Si_2_, where 0 ≤ x ≤ 0.03. The melting process was performed by using the arc-melting method under a vacuum of about 2–5 × 10^−5^ torr in an argon (Ar) atmosphere to prevent oxidation during melting. In addition, titanium (Ti) 10 g was set and initially melted before the main materials to remove the residual oxygen inside the melting chamber. To get an ingot with a homogeneous material distribution, it was flipped and remelted three times. The numerical control (NC) wire-cutting machine (EC-3025, Makino) was then used to slice the ingots into small pieces (sample’s size W × L × T = 7 × 7 × 1.5 mm) to facilitate the characterization of TE properties. The pieces were then polished in order to remove the oxidized surface before the heat-treatment process. The metallic *ε* and *α-*phases were formed during the arc-melting process. In order to transform into the *β*-phase, the heat-treatment process at 1423 K for 3 h, and consequently, the annealing process at 1113 K for 20 h, were applied for all samples in vacuumed silica quartz ampule. The first step of heat treatment was to additionally homogenize the material distribution, and the second step was to transform it into a single *β*-phase. The heat treatment and annealing process followed that of reference [23], where the optimum condition was reported.

### 2.2. Sample Characterization

The CuK*α* high-resolution X-ray diffractometer (SmartLab, Rigaku, Tokyo, Japan) was used for the powder X-ray diffraction (XRD) measurements. With Rietveld analysis utilizing the RIETAN-FP software, calculation of the crystal structure parameters and phase identification were carried out by using the measured XRD data. A scanning electron microscope (VE-8800, KEYENCE, Osaka, Japan) apparatus was then used to observe the surface structure of each of the fabricated materials. The elemental analysis was performed with a scanning electron microscope (SU8010, Hitachi High-Technologies, Tokyo, Japan) equipped with a Bruker EDS XFlash5060FQ detector. The Archimedes method was performed to measure the relative density with a gravity measurement kit (SMK-401, SHIMADZU Co., Kyoto, Japan). ResiTest8300 (TOYO Co., Aichi-ken, Japan) apparatus was used to measure mobility (*μ*_H_) and carrier density (*n*_H_) at room temperature. In addition, the electrical resistivity (*ρ*) and Seebeck coefficient (*S*) were also measured by using the ResiTest8300 at temperatures of 80–395 K and by homemade apparatus under an Ar atmosphere at temperatures of 400–800 K. The thermal conductivity (*κ*_total_) was measured by using a power efficiency measurement (PEM-2, ULVAC, Inc., Kanagawa, Japan) system and the *ZT* can be calculated by *ZT* = *S*^2^*T*/(*ρκ*_total_).

## 3. Results and Discussions

Figure 1 shows the X-ray diffraction (XRD) peaks of Fe_x_Ni_1−x_Si_2_ (0 ≤ x ≤ 0.03) at 300 K within the angles of 20° ≤ 2θ ≤ 90°. Mainly, the *β*-phase was achieved for 0 ≤ x ≤ 0.03; however, a trace of the *ε*-phase still remained at 2θ ≈ 45.2° on the right of the indexed peak (421), as zoomed in on in the inset of Figure 1. The intensity of this *ε*-phase peak increases with increasing Ni concentration from 0 to 0.03; the low intensity occurred at x ≤ 0.005. The XRD peaks of our 0 ≤ x ≤ 0.005 samples are similar to those of the study of Dąbrowski et al., who reported that a single β-phase was obtained by doping with other impurities, such as aluminum (Al) and phosphorus (P) [26]. Therefore, it is considered that the 0 ≤ x ≤ 0.005 samples had very small amounts of the *ε*-phase.

Figure 2 shows the SEM images of Fe_1−x_Ni_x_Si_2_ (0 ≤ x ≤ 0.03) captured at room temperature. The identification of phase transition by using the SEM micrograph can also be found in the previous reports of Dąbrowski et al. [26,31]. Figure 2a shows that before heat treatment, the *ε* and *α*-phases were formed at x = 0 (the bright grain represents the *ε*-phase and the dark grain represents *α*-phase). The white dots are not the microstructures but merely dust contaminated by the polished substrate. After heat treatment, for 0 ≤ x ≤ 0.005, the samples were grown in a single *β*-phase, as shown in Figure 2b–d, and for 0.01 ≤ x ≤ 0.03, the samples were grown with the majority of the *β*-phase and the minority of the *ε*-phase, as shown in Figure 2e–g. It is observed that the area or amount of the *ε*-phase increases with increasing Ni addition (x).

To observe the Ni distribution in each phase, the SEM-EDS measurement for elemental analysis was performed for the 0.005 ≤ x ≤ 0.03 sample after the heat treatment. As shown in the color mapping of Figure 3, Ni was homogenously distributed for x = 0.005 due to the formation of a single *β*-phase, whereas the Ni-richness was distributed in the area of the *ε*-phase for 0.01 ≤ x ≤ 0.03, as can be seen in the green. This tendency indicates that the semiconducting phase is moderately transformed into the metallic ε-phase by increasing Ni substitution. Furthermore, a portion of the Ni concentration is accumulated in the grain boundaries between *ε*-phase and *β*-phase, probably due to the large particle size of raw material. This issue can probably be solved by ball milling, followed by fast-sintering techniques. By utilizing the ball-milling method, the particle size of Ni can be significantly reduced, and the fast-sintering techniques could help to reduce grain growth. As the particle size reduces, the Ni might more homogenously distribute, leading to simultaneously eliminating the accumulation of Ni and grain growth. In addition, Table 1 also shows the quantitative analysis of the 0.005 ≤ x ≤ 0.03 sample. In the area of the *β*-phase for all samples, the atomic concentration of Fe was approximately 1/3, and that of Si was approximately 2/3. This indicates that Fe:Si ratio is 1:2, corresponding to *β*-FeSi_2_. On the other hand, in the area of the ε-phase, the atomic concentration of Fe was approximately ½, and that of Si was also 1/2. This indicates that the Fe:Si ratio is about 1:1, corresponding to *ε*-FeSi. In the *β*-phase area of the 0.005 ≤ x ≤ 0.03 sample, the actual Ni composition ranged from 0.003(1) to 0.010(4), indicating that the solid solution limit of Ni for *β*-FeSi_2_ is lower than x = 0.01. When the value is higher than 0.01, it facilitates the formation of the *ε*-phase. As a result, a single *β*-phase could be obtained in the 0 ≤ x ≤ 0.005 samples, as verified with the SEM image in Figure 2b–d. Moreover, as shown in Table 1, Ni in both *β* and *ε*-phases linearly increases with x, but the slope of the *ε*-phase is around six times that of the *β*-phase.

Moreover, Figure 2 shows that pore size after heat treatment is larger than that before heat treatment. This enlargement of pore size happens when the volume *β*-FeSi_2_ occupies varies with the volumes of metallic *ε* and *α*-phases during the heat-treatment process (*ε*-FeSi + *α*-Fe_2_Si_5_ → *β*-FeSi_2_). However, the relative densities range from 95.6(1)–98.7(1)%, as shown in Table 2. These values are as high as for a sample prepared by hot-pressing (HP) techniques [32], but are relatively higher than those for samples prepared by pulse plasma sintering (PPS) [28] or spark plasma sintering (SPS) [33]. This result suggests that the proposed arc-melting and direct-heat-treatment method is efficient at fabricating a high-relative-density sample that contributes positively to the decrease in electrical resistivity, which is good for TE application. The three dimensions of the orthorhombic crystal structure of *β*-FeSi_2_ were provided by our previous report [20].

The bonds of the Fe1 and sites are formed geometrically in eight coordinates, four each to Si1 and Si2, and the bonds of Fe–Si vary in length from 2.361(5) to 2.402(6) Å and from 2.282(5) to 2.415(4) Å, respectively.

The Rietveld analysis of *β*-Fe_0.995_Ni_0.005_Si_2_ after heat treatment is shown in Figure A1 (Appendix A). The calculated data, experimental data, and difference between the data, are represented by green, red, and blue lines, respectively. According to the analysis, it is considered that after the process of heat treatment, the sample is successfully grown in the *β*-phase with a trace of the metallic *ε*-phase.

Therefore, the result of the Rietveld analysis agrees with that of the SEM image. The orthorhombic structure (*Cmce* space group) was chosen for Rietveld analysis. As Ni was partially substituted into the Fe sites, 1−x was assigned as the occupied rate of Fe1 and Fe2, and x was assigned as the occupied rate of Ni1 and Ni2. In addition, both the Fe site and Si site were assigned with the isotropic atomic displacement *B* with the value of 1.0 Å^2^. A split pseudo-Voigt function was used to fit the Bragg peak profiles. A summary of the structural parameters, which were calculated by the Rietveld analysis, is reported in Table A1 (Appendix B). The lattice constants (*a*, *b*, *c*), interact atomic distances (Si-Fe), interacting atomic angles (Fe-Si-Fe), and reliability factor for weight diffraction patterns (*R*_wp_) with x dependences are plotted in Figure 4. In Figure 4a, the variations in lattice constants *a*, *b*, and *c* with x are negligible. In addition, as shown in Table A1, the change in unit-cell volume (*V*) is almost within the error range. The effect of Ni substitutions is probably not significant for the lattice constants due to the low solubility limit of Ni. Figure 4b shows that the atomic distances of Si1-Fe1 and Si1-Fe2 tend to slightly rise with increasing x, though there is no significant change in Si2-Fe1 or Si2-Fe2 with x. In addition, the interactive atomic angles of both Fe1-Si1-Fe1 and Fe2-Si2-Fe2 slightly rise with x, but those of both Fe1-Si2-Fe1 and Fe1-Si2-Fe1 slightly decline as x increases, as shown in Figure 4c. It is considered that both Fe1 and Fe2 are slightly changed with Ni addition. Therefore, the Ni population should equally occupy both Fe1 and Fe2 sites, which is similar to a previous study wherein Co was doped into the *β*-FeSi_2_ system [20,34]. Figure 4d shows the reliability factor *R*_wp_ with x dependences. The *R*_wp_ value for x = 0 is about 3.316%, indicating a good fit between the observed and computed intensities. However, as x increases, the *R*_wp_ moderately increases, probably due to the increasing amount of the ε-phase, which is verified with XRD patterns in Figure 1 and the SEM image in Figure 2.

The electrical resistivity (*ρ*) with respect to the temperature dependence of Fe_1−x_Ni_x_Si_2_ is shown in Figure 5. The *ρ* significantly decreases as Ni content increases from x = 0 to x = 0.03. The decrease in *ρ* is mainly caused by the increases in ε-phase and carrier concentration (*n*_H_), as shown in the inset of Figure 5. This tendency can be explained by Drude’s theory in Equation (1):*ρ* = *n*_H_^−1^|*e*|^−1^*μ*_H_^−1^
(1)
where *e* and *μ*_H_ are elementary charge and carrier mobility, respectively [35]. Equation (1) expresses that *ρ* is inversely proportional to *n*_H_. Therefore, as *n*_H_ increases, *ρ* can be effectively obtained. In Table 2, the *ρ* of the non-doped sample is 7.10 Ωcm with the *n*_H_ of only around 1.3(2) × 10^16^cm^−3^. As x increases from 0.001 to 0.03, the *ρ* remarkably decreases from 1.39 to 0.26 Ωcm due to the increase in *n*_H_ from 1.2(4) × 10^17^ to 2.3(2) × 10^18^ cm^−3^. Furthermore, the increase in the *ε*-phase with Ni substitution, as discussed for the microstructures above, should also contribute to the reduction in *ρ*. For x = 0.01, the *ρ* value of our sample was almost similar to that of the one Tani and Kido prepared by pressure-sintering techniques [30]. However, for x = 0.03, the *ρ* of our sample was about two times lower due to the higher *μ*_H_. The *μ*_H_ of our sample was 10(1) cm^2^V^−1^s^−1^, and that of their sample was only around 0.27 cm^2^V^−1^s^−1^. If we compare another dopant, cobalt (Co), at the same doping level, the Ni-substituted material has a much higher value of *ρ* than the Co-substituted materials. This is because Co has a higher solid solution limit in *β*-FeSi_2_; its value is up to 0.116, as reported by Kojima et al. [36]. In addition, Nagai et al. reported that the *ρ* of the x = 0.06 thin film was 0.076 Ωcm. Such a low value of *ρ* in a thin-film sample should be mainly affected by the large *μ*_H_. It is considered that the *ρ* of the bulk sample prepared by the arc-melting method should drastically decrease if the doping amount is up to x = 0.06 due to the increase in *n*_H_; however, the thermoelectric power will be deteriorated due to the effect of the metallic ε-phase.

The Seebeck coefficient (*S*) with temperature dependence is shown in Figure 6. The |*S*| of the non-doped sample (x = 0) remarkably decreases from about 290 μVK^−1^ to approximately 0 μVK^−1^ as the temperature increases from 420 to 800 K. It is considered that the bipolar effect dominates at high-temperature regions in pristine *β*-FeSi_2_ due to low carrier density. For the Ni-doped β-FeSi_2_ system, the *S* is positive at temperatures 80–115 K and 80–195 K for x = 0.001 and 0.03, respectively, indicating the p-type materials. At higher temperatures, the *S* becomes negative, indicating an n-type conduction material. This result is consistent with that of Tani and Kido [30], who also reported that the sign of the Hall coefficient (*R*_H_) changes from positive to negative at 160 K. It is considered that conduction is dominated by both holes and electrons, and its ratio varies depending on temperature. In addition, as x increases, the |*S*| becomes more stable from room temperature to 800 K. This tendency suggests that the bipolar effect is remarkably reduced with Ni substitution due to the increase in *n*_H_. The bipolar effect was reduced by Ni doping; however, at high temperatures, it was not completely eliminated. This is probably due to the much lower actual Ni doping concentration in the *β*-phase. The increase in *n*_H_ contributes to the reduction in |*S*|. The relationship between |*S*| and *n*_H_ can be expressed by Mott’s formula:(2)$$S=\frac{k_{B}^{2}T}{3\left|e\right|\hbar^{2}}m^{*}{\left(\frac{\pi }{3n_{H}}\right)}^{2/3}$$
where *k_B_*, *T*, *e*, *ℏ*, *m**, and *n*_H_ are Boltzmann constant, temperature, elementary charge, Planck constant, effective mass, and carrier concentration, respectively [37]. Equation (2) indicates that the |*S*| is inversely proportional to *n*_H_; therefore, as can be seen in Figure 6, for 0.001 ≤ x ≤ 0.03, the |*S*| of the Ni-doped samples decreases with x. Furthermore, the inset of Figure 6 shows that *μ*_H_ decreases with x, probably owing to the difference in the effective mass between the electron and the hole. This tendency can be expressed by Equation (3):(3)$$m^{*}=\frac{e\tau }{\mu_{H}}$$
where *e*, *τ*, and *μ*_H_ are elementary charge, scattering time, and mobility, respectively [35]. When the effective mass of the electron is larger than that of the hole, the mobility of the electron is lower. As shown in Figure A2 (Appendix A), for 0.001 ≤ x ≤ 0.03, the |*S*| decreases with *n*_H_, and the tendency of the experimental values of |*S*| fits with that of the calculated values (solid black curve, in the case of *m** = 0.1 *m*_e_) using the Mott’s formula in Equation (2). It is confirmed that Mott’s theory implies for 0.001 ≤ x ≤ 0.03. For 0 ≤ x < 0.001, the experimental value of |*S*| is out of the fitting curve; therefore, this might be possibly described by a two-carrier model [38]. The highest value of |*S*| was obtained for the x = 0.001 sample with the value of 450 μVK^−1^ at 450 K.

Figure 7 shows the power factor (*PF*) with temperature dependence. The *PF* is calculated by *PF* = *S*^2^*ρ*^−1^. The improvement in *PF* contributes positively to enhancing TE performance (*ZT*). The *PF* of the non-doped sample exhibited the highest value of around 3.5 μWm^−1^K^−2^ at around 450 K, as shown in the inset of Figure 7. By doping with Ni, the *PF* can be significantly improved; the maximum value was around 200 μWm^−1^K^−2^ at 600 K, achieved by the x = 0.001 sample. The enhancement in *PF* is caused not only by the remarkable increase in, *S* but also by the reduction in *ρ*. Compared to previous work reported by Komabayashi et al., the *PF* of thin-film x = 0.06 was 17 μWm^−1^K^−2^ at 300 K [28]. This value is similar to that of our bulk x = 0.001 sample with the *PF* of about 13 μWm^−1^K^−2^ at 300 K. The thin-film sample usually had a much lower *ρ* than that of the bulk sample, which provided a better *PF*. However, the high value of |*S*| for our bulk sample also increased the *PF*, which is comparable to that of the thin-film sample. In addition, Nagai et al. reported that the highest *PF* of the bulk x = 0.01 samples prepared by mechanical milling and hot pressing was about 50 μWm^−1^K^−2^ at 650 K [29], whereas that of our x = 0.01 sample prepared by arc-melting was about 130 μWm^−1^K^−2^ at 750 K. The higher *PF* in our sample is owed to the larger |*S*|. It is considered that our sample had a lower *ε-*phase amount as a result of the heat-treatment process, as that was not applied to their sample. This might be a reason why the |*S*| of their sample was lower. Therefore, heat treatment is necessary for the fabrication of *β*-FeSi_2_ for TE applications.

The total thermal conductivity (*κ*_total_ = *κ*_l_ + *κ*_e_, where *κ*_l_ and *κ*_e_ are the lattice and electronic thermal conductivity, respectively) of all samples is plotted in Figure 8. It is shown that the *κ*_total_ of Ni-doped samples is slightly higher than that of the on-doped one. This is probably because of the increase in the metallic ε-phase with increasing Ni content, as can be proved by XRD patterns and SEM-EDS analysis. As shown in the inset of Figure 8, the electronic thermal conductivity (*κ*_e_) increased with x due to the decrease in *ρ*. The *κ*_e_ was calculated by the Wiedemann–Franz law (*κ*_e_ = *L*_O_*T*/*ρ*, where *L*_O_ is the Lorenz number). The *L*_O_ was calculated by the measured Seebeck coefficient |*S*| in the case of acoustic phonon scattering (r = −1/2). Equation (4) explains the relationship between *L*_O_ and *r*:(4)$$L_{O}={\left(\frac{k_{B}}{e}\right)}^{2}\left[\frac{\left(r+\frac{7}{2}\right)F_{r+\frac{5}{2}}\left(\eta \right)}{\left(r+\frac{3}{2}\right)F_{r+\frac{1}{2}}\left(\eta \right)}-\left\{{\frac{\left(r+\frac{5}{2}\right)F_{r+\frac{3}{2}}\left(\eta \right)}{\left(r+\frac{3}{2}\right)F_{r+\frac{1}{2}}\left(\eta \right)}}^{2}\right\}\right]$$
where the function is given as: $F_{n}\left(\eta \right)=\int_{0}^{\infty }\frac{\chi^{n}}{1+e^{\chi -\eta }}d\chi$, $\chi =\frac{E}{k_{B}T},$ $\eta =\frac{E_{F}}{k_{B}T}$ and $E_{F}$ is Fermi energy [39]. Table 2 shows that the values of *L*_O_ increase with x for 0.001 ≤ x ≤ 0.03. This tendency shows that the *β*-phase moderately transforms into the *ε*-phase as the level of Ni doping increases. The increase in *L*_O_ also contributes to the high *κ*_e_ because of its proportionality.

The *ZT* value with temperature dependences is plotted in Figure 9. In addition, the inset of Figure 9 shows that the x = 0 sample had the highest *ZT* value of 2.6 × 10^−4^ at the temperature of 450 K. If we compare the *ZT* value of pristine material to Ni-doped materials, its value is very low. In the Ni-doped system, the maximum *ZT* of around 0.019 at 600 K was obtained in x = 0.001 owing to the enhancement of the power factor. When x is higher than 0.001, the *ZT* is decreased due to the reduction in |*S*| caused by the increased amount of metallic ε-phase. Therefore, it is considered that for Ni-doped *β*-FeSi_2_, the low doping amount, below 1%, is more effective at improving the TE properties.

## 4. Conclusions

Thermoelectric (TE) materials *β*-Fe_1−x_Ni_x_Si_2_ were fabricated for 0 ≤ x ≤ 0.03 by the conventional arc-melting method, followed by a heat treatment and annealing process. Traces of the *ε*-phase were formed for all samples; the lowest amount of it was obtained at x ≤ 0.005. The solid solution limit of Ni in the *β*-phase is below x = 0.01, and the *ε*-phase increases with increasing Ni concentration. As x increases, the electrical resistivity (*ρ*) and Seebeck coefficient |*S*| decrease, owing to the increases in ε-phase and carrier density. As a result, the optimum doping amount to achieve a maximum power factor (*PF*) of around 200 μWm^−1^K^−2^ was obtained in the x = 0.001 sample due to significant enhancement in |*S*|. The *PF* value of this sample was comparable to that of the thin-film sample reported by Komabayashi et al. [28]. However, this value is higher than that of the hot-pressed sample reported by Nagai et al. [29], resulting from the improvement in |*S*|. The improvement in *PF* led to obtaining the *ZT* of 0.019 at 600 K in the same x = 0.001 sample. It would be worth investigating a method to increase Ni’s solubility in β-FeSi_2_. As Ni solubility increases, the ε-phase can be reduced, resulting in an improvement in *S* and a decrease in thermal conductivity (*κ*). Therefore, *ZT* can be more significantly enhanced, making the material suitable for industrial waste heat recovery in mid–high temperature applications.

## Figures and Tables

**Figure 1 materials-16-00927-f001:**
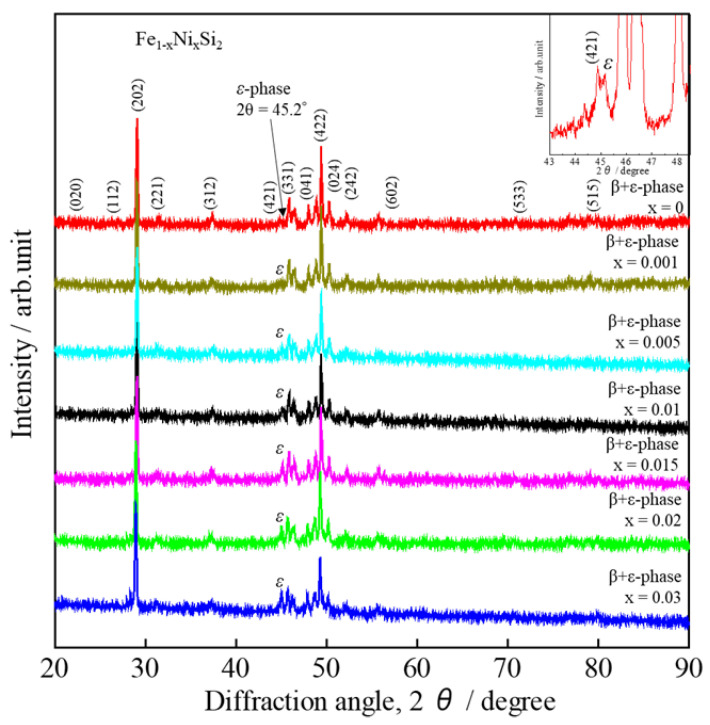
X-ray diffraction patterns of *β*-Fe_1−x_Ni_x_Si_2_ (0 ≤ x ≤ 0.03) at room temperature.

**Figure 2 materials-16-00927-f002:**
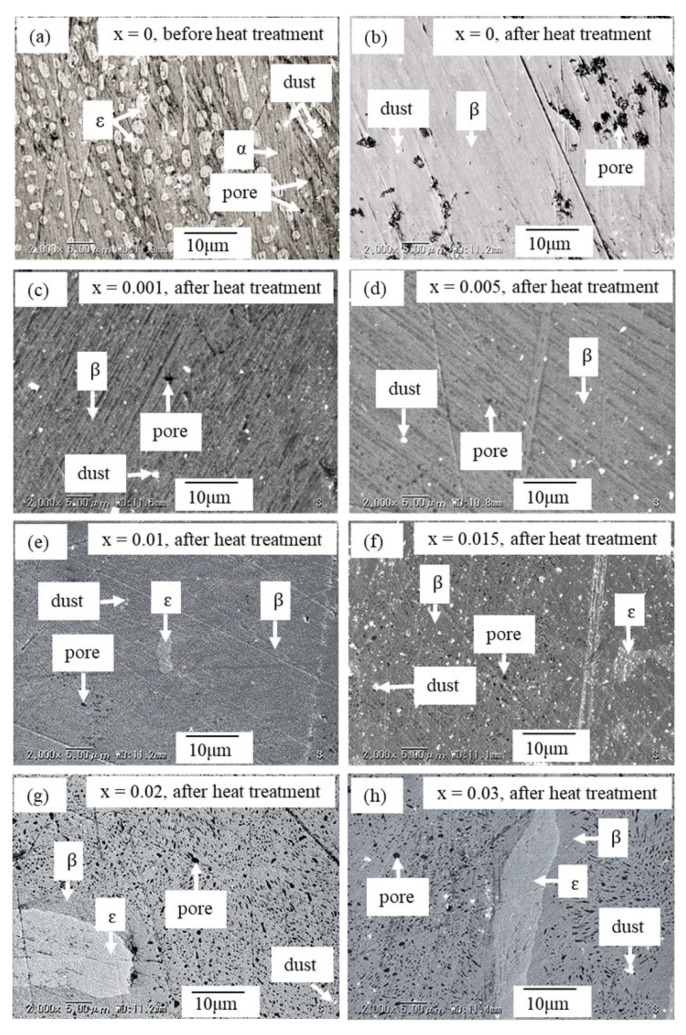
SEM image of *β*-Fe_1−x_Ni_x_Si_2_: (**a**) before heat treatment for x = 0, (**b**–**h**) after heat treatment for 0 ≤ x ≤ 0.03.

**Figure 3 materials-16-00927-f003:**
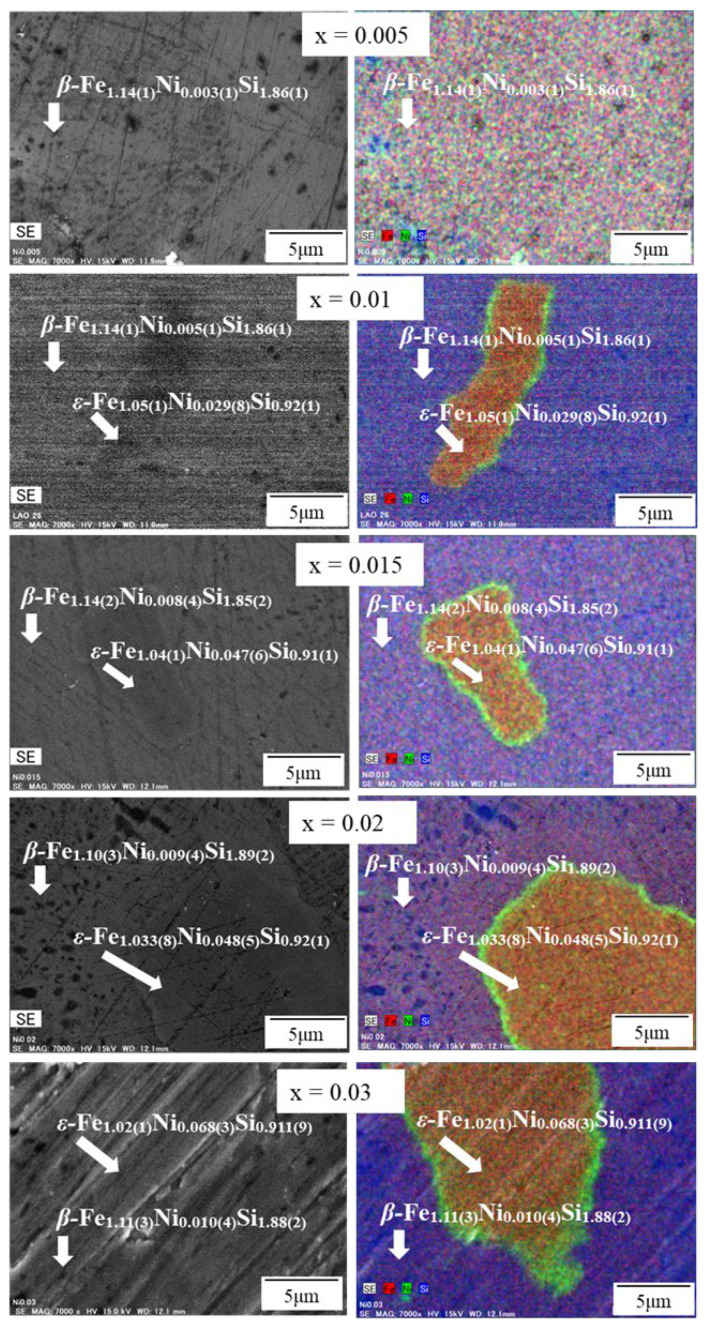
SEM-EDS mapping of *β*-Fe_1−x_Ni_x_Si_2_ (0.005 ≤ x ≤ 0.03). Fe, Ni, and Si are mapped with red, green, and blue, respectively.

**Figure 4 materials-16-00927-f004:**
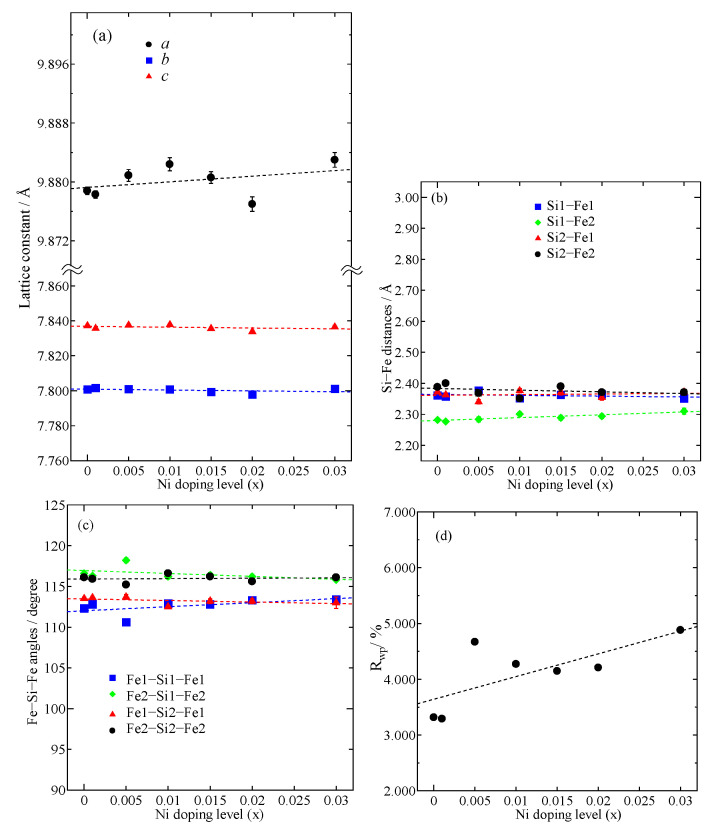
(**a**) Lattice constants *a*, *b*, *c*; (**b**) interactive atomic distances of Fe-Si, (**c**) interactive atomic angles of Fe-Si-Fe, and (**d**) reliability factor *R*_wp_ for *β*-Fe_1−x_Ni_x_Si_2_ (0 ≤ x ≤ 0.03) at room temperature.

**Figure 5 materials-16-00927-f005:**
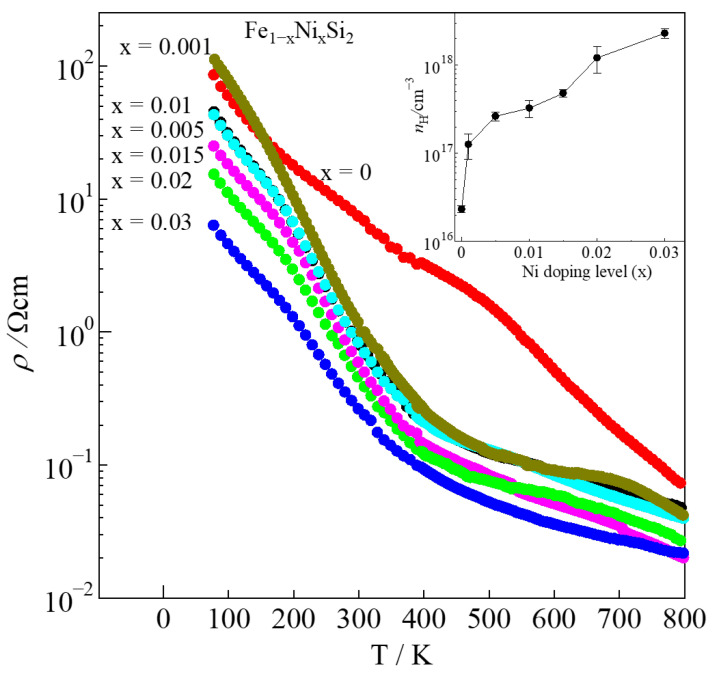
Electrical resistivity of *β*-Fe_1−x_Ni_x_Si_2_ (0 ≤ x ≤ 0.03) with respect to temperature, where the carrier concentration, *n*_H_, at 300 K is plotted in the inset.

**Figure 6 materials-16-00927-f006:**
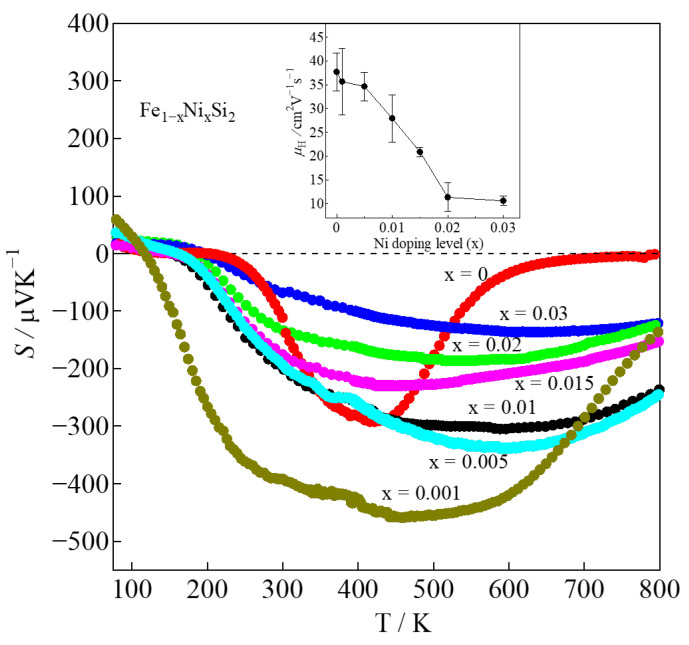
Seebeck coefficient of *β*-Fe_1−x_Ni_x_Si_2_ (0 ≤ x ≤ 0.03) with respect to temperature, where the carrier mobility, *μ*_H_, at 300 K is plotted in the inset.

**Figure 7 materials-16-00927-f007:**
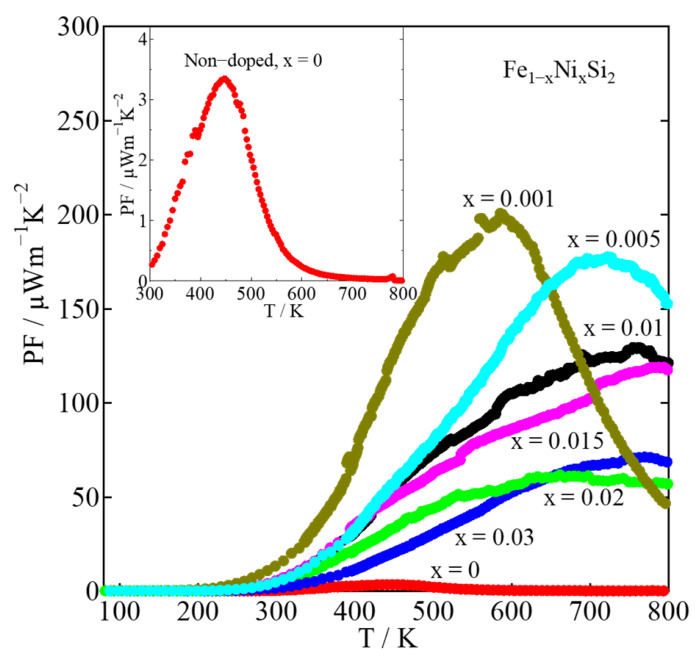
Power factor (*PF*) of *β*-Fe_1−x_Ni_x_Si_2_ (0 ≤ x ≤ 0.03) with respect to temperature, where the inset plots the data of x = 0.

**Figure 8 materials-16-00927-f008:**
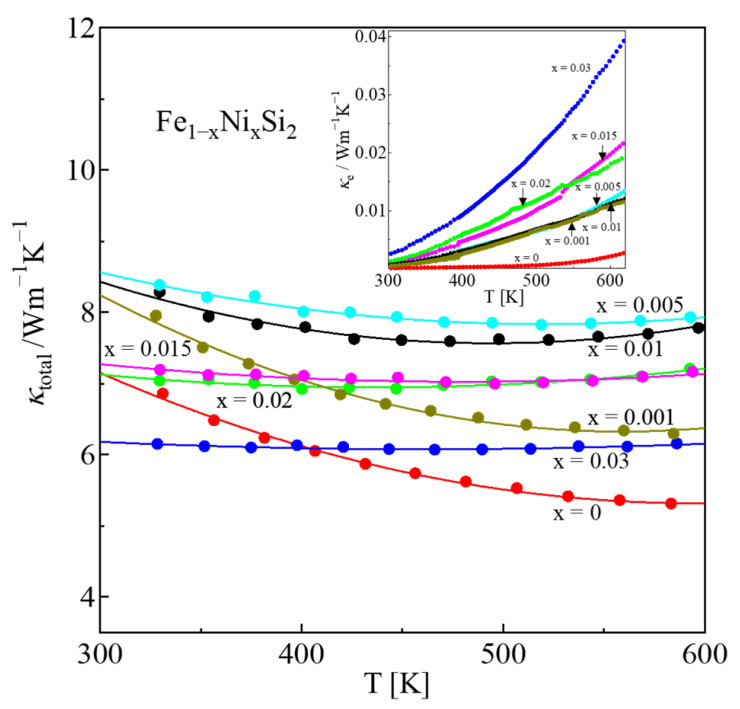
Thermal conductivity (*κ*_total_ = *κ*_L_ + *κ*_e_) of *β*-Fe_1−x_Ni_x_Si_2_ (0 ≤ x ≤ 0.03) with respect to temperature, where the inset plots the electronic thermal conductivity (*κ*_e_ = *L*_ο_*T*/*ρ*).

**Figure 9 materials-16-00927-f009:**
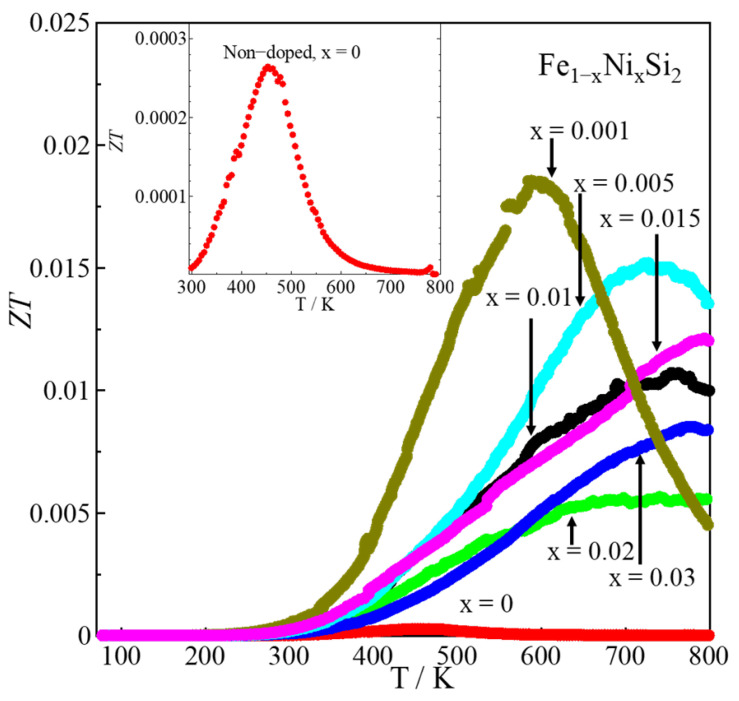
*ZT* of *β*-Fe_1−x_Ni_x_Si_2_ (0 ≤ x ≤ 0.03) with respect to temperature, where the inset plots the data for x = 0.

**Table 1 materials-16-00927-t001:** Elemental composition of *β*-Fe_1−x_Ni_x_Si_2_ (0.005 ≤ x ≤ 0.03) quantified by SEM-EDS analysis.

x	Area	Element	Atomic %	Composition Ratio	Symbol
0.005	*β*	Fe	38.1(4)	1.14(1)	*β-*Fe_1.14(1)_Ni_0.003(1)_Si_1.86(1)_
		Ni	0.10(5)	0.003(1)	
		Si	61.8(4)	1.86(1)	
0.01	*β*	Fe	37.9(5)	1.14(1)	*β*-Fe_1.14(1)_Ni_0.005(1)_Si_1.86(1)_
		Ni	0.17(4)	0.005(1)	
		Si	62.9(5)	1.86(1)	
	*ε*	Fe	52.7(4)	1.05(1)	*ε*-Fe_1.05(1)_Ni_0.029(8)_Si_0.92(1)_
		Ni	1.5(4)	0.029(8)	
		Si	45.8(6)	0.92(1)	
0.015	*β*	Fe	38.2(6)	1.14(2)	*β*-Fe_1.14(2)_Ni_0.008(4)_Si_1.85(2)_
		Ni	0.3(1)	0.008(4)	
		Si	61.5(6)	1.85(2)	
	*ε*	Fe	52.0(6)	1.04(1)	*ε*-Fe_1.04(1)_Ni_0.047(6)_Si_0.91(1)_
		Ni	2.3(3)	0.047(6)	
		Si	45.7(7)	0.91(1)	
0.02	*β*	Fe	36.8(9)	1.10(3)	*β*-Fe_1.10(3)_Ni_0.009(4)_Si_1.89(2)_
		Ni	0.3(1)	0.009(4)	
		Si	62.9(8)	1.89(2)	
	*ε*	Fe	51.7(7)	1.033(8)	*ε*-Fe_1.033(8)_Ni_0.048(5)_Si_0.92(1)_
		Ni	2.4(2)	0.048(5)	
		Si	45.9(6)	0.92(1)	
0.03	*β*	Fe	37.1(9)	1.11(3)	*β*-Fe_1.11(3)_Ni_0.010(4)_Si_1.88(2)_
		Ni	0.3(1)	0.010(4)	
		Si	62.5(8)	1.88(2)	
	*ε*	Fe	51.1(5)	1.02(1)	*ε*-Fe_1.02(1)_Ni_0.068(3)_Si_0.911(9)_
		Ni	3.4(2)	0.068(3)	
		Si	45.5(4)	0.911(9)	

**Table 2 materials-16-00927-t002:** Summary of thermoelectric properties of *β*-Fe_1−x_Ni_x_Si_2_ (0 ≤ x ≤ 0.03) at 300 K, where *L*_O_, r = −1/2, *n*_H_, *μ*_H_, *S*, *ρ*, and *κ* are Lorenz number, scattering factor (for acoustic phonon scattering), carrier density, mobility, Seebeck coefficient, electrical resistivity, and thermal conductivity, respectively.

x	*L*_O_ [V^2^K^−2^]	r	*n*_H_ [cm^−3^]	*μ*_H_ [cm^2^V^−1^s^−1^]	|*S*| [μVK^−1^]	*ρ* [Ωcm]	*κ* [Wm^−1^K^−1^]	Relative Density [%]
0	1.792 × 10^−8^	−1/2	2.3(2) × 10^16^	37(4)	127	7.10	7.16	98.0(1)
0.001	1.624 × 10^−8^	−1/2	1.2(4) × 10^17^	35(7)	393	1.39	8.25	98.3(1)
0.005	1.656 × 10^−8^	−1/2	2.6(2) × 10^17^	34(3)	194	0.69	8.57	96.24(8)
0.01	1.648 × 10^−8^	−1/2	3.2(7) × 10^17^	27(5)	205	0.69	8.44	97.5(3)
0.015	1.674 × 10^−8^	−1/2	4.8(4) × 10^17^	20(1)	176	0.63	7.27	98.7(1)
0.02	1.766 × 10^−8^	−1/2	1.2(4) × 10^18^	11(3)	135	0.45	7.12	95.6(2)
0.03	2.139 × 10^−8^	−1/2	2.3(2) × 10^18^	10(1)	62	0.26	6.18	95.6(1)

## Data Availability

Not applicable.

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
