# Peer review of "Effect of Ni Substitution on Thermoelectric Properties of Bulk β-Fe1−xNixSi2 (0 ≤ x ≤ 0.03)"

_materials, 2023, doi:10.3390/ma16030927_

Round 1

Reviewer 1 Report

The manuscript presents thermoelectric (TE) materials β-Fe1-xNixSi2 were fabricated for 0≦x≦0.03 by the conventional arc-melting method followed by heat treatment and annealing process. The thermoelectric properties are reported over the measured temperature of 80 - 800 K. At high temperatures, the Seebeck coefficient of Ni-doped samples is higher and more stable than that of the non-doped sample due to the reduction in bipolar effect. The electrical resistivity decreases with increasing x owing to the increase in metallic ε-phase and carrier density. The ε-phase increases with Ni substitution and solid solution limits of Ni in β-FeSi2 can be obtained lower than 1%. The highest power factor of 200 μWm-1K-2 at 600 K is obtained in x = 0.001, resulting in the enhanced ZT value of 0.019 at 600K. The manuscript is well organized. Thus, I recommend publication of this manuscript in Materials after minor modifications by the authors.

1.      The authors mentioned the thermoelectric properties are reported over the measured temperature of 80 - 800 K., why start with 80 K instead of room temperature?

2.      In the conclusion, the improvement in PF leads to obtaining the ZT of 0.019 at 600 K in the same x = 0.001 sample. What are the application prospects of studying the materials with such low ZT value? It is suggested to add some application prospects of the material.

3.      Please cite some references: Rare Met. (2021) 40(8):2017–2025, ACS Appl. Mater. Interfaces 2019, 11, 22457−22463, Nanoscale, 2018, 10, 15130–15163.

Author Response

Please check attached PDF file.

Reviewer 2 Report

Summary comment:

The overall intention of the paper “Effect of Ni substitution on thermoelectric properties of bulk β-2Fe1-xNixSi2” by Sopheap Sam et al., is a good one. Because of their unique electrical, magnetic and thermoelectric properties and their potential applications in many industrial fields, I agree that more attention should be paid to investigate the correlation between the structural, electrical and thermoelectric properties of compounds materials, such as Fe1-xNixSi2. These attempts help us to improve the quality of procedures used in designing materials and also to improve the thermoelectric properties of such a material.

Both language and readability of this paper are very good. Thus, the paper is very well written and is well balanced between the different experimental techniques used in the characterization of bulk β-2Fe1-xNixSi2 for thermoelectric applications. In the Introduction, the way the authors present the interventions of other studies in relation to the subject of this submission is excellent. Comparisons between these different interventions are good. The crucial findings and explanations of this submission are deep.

In my opinion this paper is worth to be published in Materials journal.

Some minor revisions for the authors to consider:

Comment 1:

·         Line 47: Unfavorable for TE application>>>> Which is unfavorable for TE application

·         Line 166: Fig.2(a)>>>Figure 2 (a)

·         Line 248: Fig. 5 (d)>>> Figure 4 (d)

Comment 2: All equations used in this work must be referenced. 

Comment 3: The parameters that appear in both equations 2 and 3 must be defined.

Comment 4: The Quality of Figure 4 is required.

Author Response

Please check attached PDF file.

Reviewer 3 Report

The paper investigates the effect of Ni substitution on thermoelectric properties of bulk β-phase of FeSi2, in the range of 0≦x≦0.03. It is a comprehensive and easy to read paper, which seeks to improve the thermoelectric performance of materials with poor efficiency, like FeSi2. It is an interesting attempt for cheaper and less toxic materials, sacrificing the performance of considerably higher performing thermoelectrics. Authors should make in the introduction, a clear point on the benefits of this approach vs. the performance sacrifice.

Some additional points that authors should take into account to improve their manuscript:

1.     Not all of the inserted Ni acts as a dopant in β-FeSi2 phase and ε-FeSi phase. As can be seen in Figure 3, a portion of the nominal Ni concentration is accumulated in the boundaries of the ε-phase. Authors should make a remark on that, suggesting potential ways to eliminate it.

2.     Furthermore, as can be seen from Table 1, Ni in both β- and ε-phases follow a linear trend on x. The slope on ε-phase is ~ 6 times higher than in the β-phase. Authors should make a remark on that as well.

3.     Authors claim that they have achieved a reduction of the bipolar effect in FeSi2. However, the upturn in the Seebeck coefficient in Figure 6, at high temperatures is indicative of the appearance of bipolar effect. The bipolar effect has been reduced, however not completely eliminated, and this may be due to the much lower actual doping concentration in the β-phase. Authors should make a remark on that as well.  

4.     Authors mention in line 300 that “the relationship between |S| and nH can be expressed by Mott’s formula”. However, the Mott’s formula holds for highly degenerated electron gas, which seems not to be the case in this work. Furthermore, the Mott’s formula implies a monotonous decrease of |S| with the increase on nH, which is not the case for undoped (x=0) FeSi2 and Fe1-XNiXSe2 (x=0.001), as can be seen in Table 2. I believe that the “anomalous” behavior (both |S| and nH increase simultaneously) can be described by a two-carrier model. Authors should make a remark on that as well. A Pisarenko-like plot, taking into account the two (conduction & valence) bands might be helpful.

5.     Finally, a careful proof-reading of the manuscript, by a native English speaker, would eliminate the translation errors, such as “stable” instead of “uniform?” in line 16, “relevant” instead of “suitable” in line40, “stability” instead “monotonicity” in line 56, “larger” instead of “more” in line 91, etc. These are some examples.

Author Response

Please check attached PDF file.
